# Women's lived experiences of sex hormones and life-related to bariatric surgery: an interpretative qualitative study

Rebecca Paul [1,2] Ellen Andersson,[2] Torsten Olbers,[2,3] Jessica Frisk,[2]
Carina M Berterö [4]

[1]Department of Surgery, Falun County Hospital, Center for Clinical Research Dalarna, Falun, Sweden
[2]Department of Surgery and Department of Biomedical and Clinical Sciences, Linköping University, Norrköping, Sweden
[3]Wallenberg Centre for Molecular Medicine, Department of Biomedical and Clinical Sciences, Linköping University, Linkoping, Sweden
[4]Division of Nursing Sciences and Reproductive Health, Institution of Medical and Health Sciences, Linköping University, Linkoping, Sweden

**Correspondence to**
Dr Rebecca Paul;
rebecca.paul@liu.se

## ABSTRACT

**Objectives** The study aimed to explore the lived experiences of women with severe obesity before and after undergoing bariatric surgery with a special focus on possible effects of changed sex hormone levels.

**Design** A qualitative interview study with transcribed text analysis based on Gadamer's hermeneutics.

**Setting** Regional hospital and outpatient bariatric clinic in central Sweden.

**Participants** Ten women (age 23–38 years) having undergone Roux-en-Y gastric bypass surgery between 2016 and 2019 were interviewed.

**Results** The transcribed interviews were analysed according to Gadamer's hermeneutics. Text horizons, interpreter horizons and fact horizons were derived and formed the fusions 'Recognition of unhealthy body weight', 'Dealing with other people's opinions and society's norms', 'Life has changed in a positive way' and 'Accepting inner self and bodily changes'.

**Conclusion** Women highlighted weight and body size in their responses. The study provided a deeper understanding of the situation of women living with obesity and pros and cons of having undergone bariatric surgery. Experiences of changes in sex hormones and fertility were discussed but not central to the informants. Participants emphasised the need to be prepared and properly supported in dealing with changes in life after bariatric surgery and subsequent weight loss.

## INTRODUCTION

Obesity is a global health issue that has reached epidemic proportions. In 2017, the WHO reported that overweight and obesity led to 4 million deaths worldwide.[1] Obesity is associated with an increased risk of developing comorbidities that can lower quality of life, decrease physical function and shorten life span. Medical consequences of having obesity include cardiovascular diseases, diabetes, musculoskeletal disorders, certain forms of cancer and effects on sex hormone regulation and fertility.[2 3]

Altered sex hormone regulation in women with obesity can lead to subfertility, perinatal

---

## STRENGTHS AND LIMITATIONS OF THIS STUDY

⇒ Interview study that allows for an in-depth understanding of the explored situation.

⇒ Gadamer's hermeneutics allows for an interpretation of the rich text data with consideration taken to the interpreter's pre-understanding as well as current facts related to the phenomena presented by the interviewed women.

⇒ The interviewed women prioritised a focus on weight and body size and the descriptions concerning experiences of changed hormones were not as emphasised as expected by the research team despite specific questions concerning hormones.

⇒ The use of hormone-based contraceptives may have been a confounding factor influencing experiences in changes of hormones in some women.

⇒ Language limitations may have reduced the detail in the women's descriptions as only Swedish speaking participants were included in the study.

---

complications, as well as bodily changes secondary to hyperandrogenism.[4 5] Effects of obesity on fertility are irregular periods, reduced number of viable egg cells and disrupted implantation.[6] Pregnant women with obesity are at increased risk of miscarriage, gestational diabetes and pre-eclampsia as well as large for gestational age babies, congenital defects and birth complications such as haemorrhage.[7]

The disrupted female sex hormone balance leads to male-pattern fat dispersion with central fat accumulation, as well as hirsutism with increased body hair, especially in the facial area. These hormonal consequences lead to the loss of female body form and, in conjunction with irregular periods and subfertility, lead to an experience of lost female characteristics.[8] A recent study[9] has illustrated that women living with obesity have reduced self-esteem and increased body image dissatisfaction perceived as due to their weight and

body shape. The effects on self-esteem and body image are also associated with higher incidence of depression influencing psychosocial health and well-being.

Bariatric surgery, such as Roux-en-Y gastric bypass (RYGB) surgery, is a well-proven treatment leading to sustained weight loss and improvements in comorbidities in patients with severe obesity.[10] Previous studies have found a reversal of disrupted sex hormone levels in women and improved fertility after RYGB surgery.[11 12] Other studies show that women experience several physical changes after bariatric surgery with a return to a more female pattern of fat and hair dispersion.[13 14]

Several quantitative studies using questionnaires have been conducted to investigate health and quality of life after bariatric surgery. Results suggest that physical components of quality of life improve and remain so over time after bariatric surgery while short-term improvements in mental components of quality of life during the first year are followed by a long-term erosion to preoperative levels.[15 16] Data on mental health (ie, depression, anxiety, self-harm) after bariatric surgery are conflicting.[17] However, although high body mass index-levels are associated with an increased prevalence of mental health challenges there does not seem to be any long-term improvement in mental health secondary to weight loss.[18]

Qualitative studies have investigated the experiences of patients living with obesity[19 20] and after undergoing bariatric surgery.[21 22] However, to our knowledge, no qualitative studies have been conducted to explore women's encounters of hormonal changes after bariatric surgery.

This qualitative study aims to explore the lived experiences of fertile women with obesity prior to and after undergoing laparoscopic RYGB surgery with an intended focus on experiences related to hormonal changes after surgery.

## METHODS

Gadamer's hermeneutic analysis was chosen in order to achieve a richer and deeper understanding of the experiences described by the women included in the study.[23] Gadamer's hermeneutics involves creating horizons from the participant's text, the investigators interpretation of the participant's text and supportive text from current research concerning the isolated participant text excerpts. Creating a whole broader meaning out of individual narratives while taking into consideration the pre-understanding of the interpreter and the current accepted fact of the time is a hallmark of hermeneutic technique. Current accepted facts involve reviewing and including relevant contemporary studies supporting the phenomena described by the participants and merging these findings into the derived fusions.

## Sample

Purposive sampling strategy was used and women considered to be of fertile age, 18–40 years, that had previously undergone RYGB surgery between 2016 and 2019 were the focus of the study sample. The time interval of surgery was chosen for a recent memory of life before surgery while allowing for experience of changes after surgery. Exclusion factors were women who do not speak Swedish or were otherwise prevented from participating in an interview such as having difficulty in understanding questions or expressing themselves. Communication via interpreter was avoided due to the risk of information distortion through a third party and the risk of questions and responses being misunderstood. Thirty women fitting the inclusion and exclusion criteria were identified from the bariatric surgery outpatient clinic in Falun County Sweden, and contact was made by sending information per post indicating that the first author would be communicating with the women within a few weeks to consider participation in the study. Women were provided a telephone number and an email to contact in case they desired to opt-out and did not want any further contact. Study participant information containing a consent form was included in the posted information. Twelve women initially accepted participation. However, 2 women later declined due to family illnesses and 10 women (age 23–38 years) were subsequently interviewed. A number of 10 was deemed to be an appropriate sample size due to the interpretative qualitative method used in this study, and as previously described by Kvale and Brinkmann.[24]

## Interview

Participants were allowed to choose the environment that they found most comfortable to be interviewed in (six in the home, one at hospital conference room, one at local library study room, two per telephone). Interviews were carried out by a single trained qualitative researcher (RP) as conversations with support of an interview guide with open-ended questions. The interview guide was based on the study aim although open-ended and general terms were used to avoid influencing the women's responses (box 1). Interviews were recorded using a digital pocket recorder and transcribed verbatim using a word processing programme. The interviews lasted a median of 50 min, range 32–62 min. Interview transcripts were reviewed, and interview techniques were discussed

---

**Box 1   Interview guide (original language: Swedish)**

1. Tell us how your life has changed since undergoing bariatric surgery? (social, individual)
2. Tell me, how has your body changed since the surgery? (appearance, function, experience)
3. Tell me how your body functions since the surgery? (weight-wise, stamina, size, hormonal)
4. How do you feel that hormones affect your body? (hormones)

Follow-up questions:
1. Can you tell us more about this?
2. How has this affected you?
3. Can you elaborate on this?
4. Tell me more …
5. How has this turned out since the surgery?

---

**Table 1** Example of data analysis according to Gadamer's hermeneutics

| Horizon of text (participant) | Horizon of researcher | Horizon of literature/facts | Comments | Fusion of the horizons |
|---|---|---|---|---|
| Yes, eeeeh, physically of course, I've lost 50 kilos, it's nothing that I've noticed, like, it's when you em, partly it's when I buy new clothes, but I still don't really understand, I think I look about the same now as I did then | ▶ Not aware that looks have changed concerning weight loss<br>▶ Weight loss becomes tangible by buying clothing | ▶ Increased body dysmorphia and 'ghost fat' in the time period directly after surgery | ▶ Mind and body conscientiousness<br>▶ Body awareness after weight loss<br>▶ Takes time before it becomes clear that body and appearance have changed and it was through buying clothing that it became more obvious | Accepting inner self and bodily changes |

between the interviewer (RP) and a qualitative researcher (CMB). RP and CMB reviewed the transcribed text and determined that data detail and richness were sufficient to carry out analysis without further interviews.

## Analysis

The four principles of Gadamer's hermeneutic analysis[23 25] were used during data analysis and interpretation: (1) data immersion was carried out by reading through the transcripts repeatedly creating an understanding of the text as a whole; (2) an analysis of the text was carried out with meaningful excerpts of text from the transcripts being extracted and discussed (text horizon); (3) the interpreter's translation and understanding of the text excerpt was described and deliberated through written text (interpreter horizon) and thereafter findings from current accepted studies and literature about the existing meaningful text were presented (fact horizon); (4) horizons were merged into fusions presented in a whole text providing a broader and richer understanding of the phenomena exposed from the women's experiences. The whole study proceeded as a hermeneutic circle, from the part to the whole and back again, woman by woman, sentence by sentence. See table 1 for an example of analysis.

The first and the last author (RP and CMB, respectively) read the transcripts and had a continuous dialogue and review throughout the analysis process.

## Patients and public involvement statement

Patients were not directly involved in the planning, design or recruiting process of the study. Previous concerns posed by patients in clinical experience and previous studies (methods of recruitment, sensitivity to having obesity, comfortable interview situation) were taken into account in the development of the project design.[12 26] The results will be shared with the patients after acceptance with an emailed summary of the study.

## RESULTS

The analysis was performed with Gadamer's hermeneutics involving integration of the text horizon (text constructions given by patients), interpreter horizons (our interpretation of situations explained in interviews) and fact horizons (based on current selected literature). Merging of the horizons created the fusions 'Recognition of unhealthy body weight', 'Dealing with other people's opinions and society's norms', 'Life has changed in a positive way' and 'Accepting inner self and bodily changes' (see figure 1).

### Recognition of unhealthy body weight

The women expressed a realisation that they previously were living with unhealthy weight and needed to make the necessary decisions to do something to change their situation and reduce the risks of comorbidities related to obesity. They felt hopelessness in that their weight always ended up increasing after dieting and that they often regained more weight than was lost. This inability to maintain a healthy weight after repeated weight loss led to feelings of shame and ineptitude where they felt that they were always blamed for having obesity.

> I had a very hard time losing weight on my own, I tried a lot of different things, and it was hard, I worked out, ate GI, did weight watchers, everything, and only lost a few pounds, and then when I stopped it, I went up the double. (W1)

> It was mostly disappointment and anger at myself like how I could have done this to myself, how could I do this to my body. (W4)

The women described joint, back and foot pain as well as constant exhaustion from carrying the extra weight that kept them from exercising, hindered them from taking part in employment and isolated them from the outside world. Their health problems became evident when they developed comorbidities to obesity and these consequences made it clear to them that having obesity was a threat to their health and that they needed to seek help for a more effective weight loss.

> It felt terrible, so I was sad, really, I felt bad like … you realized that this is not good either and I was not old, I was 25 years old and could not run up a hill. (W3)

The women feared that adverse effects on health from having obesity would threaten their abilities to parent by shortening their life span and diminishing capability to be physically active with their children.

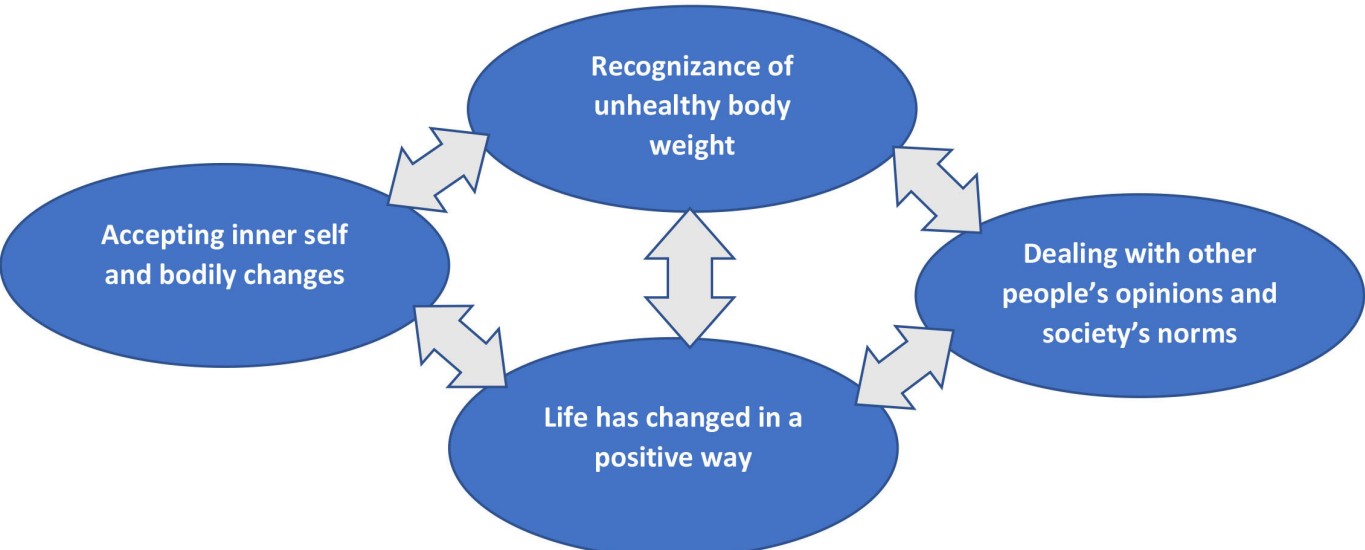

**Figure 1** Visualise the fusions and their relationship.

Now my excess weight will eventually affect my heart and I have a child who was 4 at the time. Who will take care of her if I have a heart attack or stroke and die and disappear? I want to see her grow up and that's what made me do the surgery, that was the last straw. (W2)

But it was also that the children said, "but mother why are you so tired, why don't you want to do anything, and you are boring, and you never want to do anything", and you heard it all the time. (W4)

Several of the women described difficulties in becoming pregnant before surgery and that they were aware that having obesity could lead to subfertility and reproduction-related difficulties. They feared that having obesity could hinder them from being able to have children of their own.

We tried for several years actively (to conceive) and it did not work, by itself, so we had to seek help in the end. However, I knew that the first thing they will say is that you both need to lose weight. (W5)

### Dealing with other people's opinions and society norms

The women expressed how dealing with other people's opinions of them and their weight affected the way that they felt about themselves. They did not want to be around other people and they felt that the stigma associated with being overweight made them less valued. The lack of knowledge concerning the causes of obesity among the public led the women to feel shame for their situation, weight and appearance.

I thought that nothing was fun when I was big, I didn't want to go out, I didn't want to meet people, I kept to myself a lot, because I felt that no, I had no business being around others, I didn't feel nice enough to be around other people. (W2)

I have knowledge about my operation, about me, so it's probably very difficult for others who haven't had to be in this, in what it is to have obesity, and how it is to have surgery, and the knowledge is very low among others … including healthcare personnel. (W8)

Many of the women presented their experiences of being singled out for having obesity. The women felt shame and worthlessness from their body size and evaded unwanted attention from others because they felt that the norms of society determined a person's value based on how attractive they are perceived to be in a social group.

It was mostly that I had the feeling that I was being watched when I was out among people, and whether I was in a galleria or in town, just going out and having a coffee with a friend, I felt that people were staring at me because I was so big, uh, and therefore, I withdrew to stay away from being among people. (W4)

Being fat is ugly, you've been taught that since you were little by society, that you're outside the norm, and that's probably what's ingrained in your mind, that you're a little less worthy, so to speak. (W5)

Social media's influence on stigma was expressed repeatedly by the interviewed women. The harsh and mean-spirited comments from anonymous persons could have serious effects on the women's self-esteem and mental health and led to them feeling even more isolated from society.

Post a picture on social media, and people comment on whether you looked good or not … I think there was a lot of cyberbullying, like back then, if you didn't look good you were told so …. a lot of that started then … it probably started in my generation. (W7)

The women feared that media's influence on body image could affect the way their children perceived

themselves or were treated and they expressed a need for more knowledge concerning obesity and bariatric surgery in order to reduce stereotypes.

> I think that it's hard to change with people, actually, I can see as well, um, one of my children, who is not at all overweight, at all, has expressed now quite recently that she's not comfortable with her body, and that she feels fat. (W8)

### Life has changed in a positive way

The women that were interviewed expressed that surgery led to their lives being changed in a positive way. The decreased weight burden led to increased energy and improved mobility with greater endurance for physical activity which allowed for more healthy lifestyles as well as feeling more motivated to be socially active.

> I think I'm happier as a person, more grateful, I'm trying to, like, take advantage of life, I'm going to, like, take care of myself more with both diet and exercise, I'm grateful that I got this chance. (W1)

Improved endurance and mobility with reduced musculoskeletal pain allowed the women to be employable again and improved their economic situation, including potential for career advancement. Often, there had been several failed attempts previously to rehabilitate to work due to exhaustion and physical pain.

> It made a big difference at the workplace because I had back pain. That is the most important thing about having the operation, is that I can live a life again. (W9)

Family life and relationships with their children were enriched after the weight loss from surgery. The women talked about how increased energy and mobility made it possible for them to be the parents they envisioned by playing with their children and taking part in games, outdoor activities and visits to amusement parks without fear of difficulties from their body size.

> Everything has become so much easier since I got rid of the extra kilos that were there and were bothering me before. Everything has become a lot easier, yes, I was able to socialize and activate my child in a completely different way too. (W2)

The women explained the importance of clothing and how being able to fit into the current trendy styles included them into society as well as the ability to share clothing with others bringing feelings of fellowship and solidarity. The reduced clothing size after surgery was a reminder of their successful weight loss and a motivator to continue their healthy lifestyles.

> It is so clear that it is a boost, that you can choose more clothing and any clothing store, that you are part of society.

> You don't feel like the elephant in the room anymore, you can go to a normal store and buy clothes, so it's changed a lot. (W7)

The women described how male-pattern fat dispersion dissipated after surgery and that they experienced more distinct female contours with a defined waist and hips. They felt more confident having more feminine characteristics and enjoyed that clothing fit their figures in a more complementary way. Restoration of menstrual cycles and decreased intensity of menstrual symptoms, such as pain and bleeding, was experienced by a few participants.

> At my heaviest my body shape very square, and no waist and not very much shape at all, hips and shoulders, everything was about the same width, and that's something that I noticed … that I had more defined body parts, so it's a change, purely in terms of appearance. (W6)

> I think it was six months, seven months and then the period came back, and it was really a sign that the system is working, so, so that, after a year had passed, at an annual checkup we were told that now it was ok like, that now we could try to get pregnant. (W10)

> I used to have quite a heavy period, had a lot of period pain, and things like that, and it has decreased and gotten better, I don't have as much pain, everything has become a little easier. (W7)

Sex lives were improved after weight loss and the women explained how the increase in energy, reduced body size and enhanced self-confidence increased their sex drive and motivation for taking part in sexual activities. The women described that they no longer found it uncomfortable to show their bodies without clothing.

> I get turned on to things more easily, it's not the same, before it was like I went back and thought no… I don't want to … so there have been some changes now, that the body reacts differently than what it did before … and it has, it's nice in a way, because before I thought there was something wrong with me. (W7)

### Accepting inner self and bodily changes

The women experienced many physical alterations after surgery and felt it was difficult to understand, adjust to and accept everything that occurred which led to a feeling of their mind not keeping up with the physical changes in their bodies. They felt that they were still the same size as before their surgery and were often only reminded when they purchased clothing and had to buy much smaller sizes.

> The brain doesn't react with what's happening, you don't stand in front of the mirror all the time and keep track, at least not me, the brain just doesn't keep up, that everything changes. (W9)

> The head did not keep up with the body, so it takes longer than the body to keep up with the journey. (W4)

It was important for the interviewed women to maintain the same personality and way of thinking as before

the surgery with the importance of not allowing weight loss and a changed exterior to influence their opinions of themselves. The women reacted strongly to the more positive treatment they experienced based on their looks since they felt they were the same person on the inside.

> I have the attitude that weight has nothing to do with who I am, it has nothing to do with me, I'm still the same person whether I weigh so much or so little. (W10)

There were side effects of surgery that the women dealt with in different ways. The resulting excess skin from decreased fat tissue reminded the women of their previous weight, contributed to a continued large body size, and led to certain hinders in finding clothing that fit properly and at times difficulty with hygiene. The women talked about a tolerance for dumping symptoms and early satiety and accepted that it was part of the effectiveness of the surgery.

> You look like you weigh more than you do because you have so much excess skin. (W1)

> I get heart palpitations sometimes, so you are physically reminded that you have been careless with food, so it is a huge change that the body does not react to food in the same way at all as before the operation. Now it reacts when you eat too much, or you can feel a little like maybe you shouldn't have eaten that type of food. (W8)

## DISCUSSION

This study was designed to explore the lived experiences of women with obesity before and after undergoing bariatric surgery and possible effects of changed sex hormone levels. The principal findings reflected the recognition of an unhealthy body weight as a motivation for surgery and that dealing with society's norms and other people's opinions had a profound effect on their quality of life and mental health before surgery. The women also described the changes after surgery that had a positive influence on their daily lives as well as challenges in adapting mentally to the physical alterations after weight loss.

Although hormonal changes were not a central focus in the women's stories, they mentioned related physical alterations that could be interpreted as hormonal effects that became a part of the whole experience. For example, effects related to sex hormones such as difficulty conceiving prior to surgery as well as reduction of male pattern fat dispersion and improved sexual function after surgery and weight loss were mentioned. However, despite repeated attempts to return a focus to hormones and postoperative changes, the women emphasised preoperative challenges involving living with obesity as well as postoperative benefits concerning improved physical function, quality of life as well as adapting to changes appearances.

Prior to surgery, adverse social experiences based on body size led to isolation and avoidance of social contacts potentially increasing the risk of developing mental health issues, such as depression and anxiety.[27] Stigmatising attitudes in society have been found in previous studies to have a basis in a lack of knowledge about the pathogenesis of obesity, as well as believing that obesity is associated with negative stereotypes such as laziness, lower intelligence, lack of motivation and self-control.[28] The existence of stigma is even present within the healthcare field and can lead to the women receiving substandard care.[29]

Clothing size symbolised success and being able to wear and share trendy and standard sized garments made women feel included in a larger context. Work apparel was another aspect representing inability to 'fit in'. A previous study suggested that smaller clothing sizes signified a healthy body weight and associated with more positive self-esteem and improved body image.[30] Clothing size has also been used in previous studies as a measure of risk for developing comorbidities.[31 32] The women stated that before surgery, their larger attire was a reminder of their unhealthy weight and that reaching the desired weight and clothing size symbolised their journey from having obesity to obtaining a healthy body.

The rapid weight loss and development of excess skin after surgery led to the 'mind not following the body' corroborating previous research that reported difficulties adjusting to the new body appearance and resulting in a discordance in their identity perception.[33] Interestingly, research has shown that patients who postoperatively identified themselves as non-obese were more satisfied with their body size than those patients who had lost weight but still identified themselves as having obesity. Fortunately, women became more satisfied with their body size over time.[34] In a study by Gilmartin women who had undergone bariatric surgery emphasised difficulties in changes in body image and a need for support.[35] A meta-analysis showed difficulties in finding standard follow-up measures that take into account all factors that influence a patient's satisfaction with their body after surgery.[36] Preoperative information about expected changes and experiences of others, and support postoperatively may enable women to cope with the 'mind not following the body'.

Quantitative studies have shown restoration of sex hormone regulation in women after bariatric surgery[37] and experiences likely secondary to that were expressed by several women in this study. Satisfaction with having a defined waist, reduction of central fat dispersion as well as restitution of menstrual cycles, more profound cyclic variations in mood after surgery and better sex life were expressed. This was associated with a concrete feeling of success with surgery and a reinforcement of the health effects of weight loss on the body's function. A limitation may be that a lack of awareness about the relationship between sex

hormones and experienced symptoms may, however, have led to not catching certain aspects of hormonal changes during the interviews leading to a primary emphasis on overall benefits and physical changes after surgery. Another limitation might be the language requirement and exclusion of non-Swedish speaking women who may have enriched the detail and increased transferability of the findings.

In conclusion, this study of women's lived experiences before and after bariatric surgery provided information suggesting that issues related to changes in body size and other people's attitudes are dominating. Experiences that could be related to changes in sex hormones were mentioned but not central in the women's narratives. Findings from this study illustrate the preoperative experiences of women with obesity as well as important changes after surgery and weight loss that may require specific support. Guidance concerning preoperative patient expectations and alterations occurring postoperatively can preferably be included in the information to women seeking bariatric surgery and implemented as a routine during follow-up.

**Acknowledgements**  We would like to extend our gratitude to the bariatric clinic in Falun for assisting in the recruitment of the study participants.

**Contributors**  RP conceived the idea of conducting the study with support and supervision from CMB, TO, JF and EA. The theory, methodology and interview guide were discussed by all authors and determined with expert supervision from CMB. RP performed the interviews and the transcription and discussed and processed the data under supervision from CMB. Analysis was carried out by RP and CMB. RP drafted the manuscript which all authors revised into the final manuscript. CMB is guarantor of the study.

**Funding**  This work was supported by grant number 06000984 from the County Council of Östergötland and grant number 974932 from the County Council of Dalarna, Sweden.

**Competing interests**  TO participated in advisory boards and educational activities for Johnson & Johnson and Novo Nordisk unrelated to the submitted article, and reimbursements were directed to his academic institution. All other authors have no competing interests to declare.

**Patient and public involvement**  Patients and/or the public were not involved in the design, or conduct, or reporting, or dissemination plans of this research.

**Patient consent for publication**  Not applicable.

**Ethics approval**  The study was approved in April 2022 by the Swedish Ethical Review Authority, case number 2022-01708-01. All procedures were in accordance with the ethical standards of the 1964 Declaration of Helsinki and its later amendments. Written and verbal informed consent was obtained from all study participants. Close consideration was taken to protect identities due to the sensitive subject and the limited number of informants.

**Provenance and peer review**  Not commissioned; externally peer reviewed.

**Data availability statement**  Data are available upon reasonable request. Deidentified participant data from interview transcripts and analysis routines are available upon request.

**ORCID iDs**
Rebecca Paul http://orcid.org/0000-0001-5355-8276
Carina M Berterö http://orcid.org/0000-0003-1588-135X

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
