## [Reviewer comments · BMJ Open]

ARTICLE DETAILS

TITLE (PROVISIONAL)	Women's lived experiences of sex hormones and life related to bariatric surgery: An interpretative qualitative study
AUTHORS	Paul, Rebecca; Andersson, Ellen; Olbers, Torsten; Frisk, Jessica; Berterö, Carina

VERSION 1 – REVIEW

REVIEWER	Catherine Homer Sheffield Hallam University, Centre for HHealth and Social Care Research
REVIEW RETURNED	20-Mar-2023

GENERAL COMMENTS	The article includes fascinating quotes explaining the experiences of women living with severe obesity who have undergone bariatric surgery. I was expecting the article to include more findings on the experiences of sex changes in hormones given the emphasis on this in the introduction - which is seemingly what you wanted to discuss. However I think the quotes and experiences of women are not related strongly enough to this to be such a major part and wonder if there could be some refocus to be explicit about what the article does add not what it doesn't? There are lots of really interesting themes that have come out that support other work published from across the world and I feel some of this gets lost in what you maybe hoped to say. Some specific areas to consider below. page 4 line 14 - 'The use of hormone-based contraceptives may have been a confounding factor influencing experiences in changes of hormones in some women.' I am not sure this features in the rest of the paper enough to be brought into this section., it is very much authors assumption as apposed to what is actually presented form the data. There is no other mention of the word contraceptives in the article and would expect more relevance to this statement to be included in the article summary. Page 10 line 3 - 15 - very long sentence consider rewording. Page 19 line 10 - 'Findings can preferably be included in the information to women seeking bariatric surgery and be repeated as a routine during follow-up' - Unclear what this means - please consider rewording.
---

REVIEWER	Clare Robertson University of Aberdeen, Health Services Research Unit
-----------------	--

GENERAL COMMENTS

Thank you for the opportunity to read this interesting paper. The study provides valuable insights into the experiences of women who have undergone bariatric surgery. I have a few suggestions to, hopefully, improve the manuscript for a wider audience who may be less familiar with issues surrounding obesity and fertility/sex hormones and/or may have less experience of qualitative methods.

Introduction

Line 25: To improve the readability for a wider audience, it would be helpful to provide lay descriptions of some of the less well-known technical terms relating to fertility, such as fecundity, oligo/anovulation and oocytes – I think fertility, irregular periods and egg cells are possibly more commonly used terms?

Line 44: Can you describe the effects on women's physical and psychosocial health and well-being? I presume these are negative effects?

Methods

Line 42: Please can you provide a brief description of Gadamer's hermeneutic analysis and why you believe this was the best qualitative methodological approach to adopt to give you the "richer and deeper understanding of the experiences described by the women included in the study". Providing a fuller description of your methodological approach might address some of my other queries regarding the term 'facts' and interviewer reflexivity.

Line 49: Can you please explain the phrase "the current accepted fact of the time"? What facts are you referring to here and from whose perspective are these facts determined?

Line 10-13: Please describe the process by which women were excluded from participating in an interview due to their difficulty in understanding questions or expressing themselves. How was this criterion determined and by whom?

Line 31-33: Please expand your description of your decision for determining that ten participants was the number deemed to be an appropriate sample size. Was the sample size pre-determined before recruitment/data collection began or did you achieve data saturation with ten participants? If you achieved data saturation, please describe your method for determining data saturation.

Analysis section: You state earlier in the methods section that your methodological approach allows consideration of the pre-understanding of the interpreter. Can you provide more details of whether the researcher critically examined their own role and potential for bias during data collection and analysis and, if this was done, how this was achieved in practice?

Line 56-57: Again please explain what you mean by facts in the sentence "current facts about the existing meaningful text were presented (fact horizon)."

Patients and public involvement statement

Line 50: Can you please cite some of the previous studies that you discuss here?

	Results and discussion Given the emphasis on sex hormones in your study design, it would be helpful to have a little more detail/clarity on the women's thoughts on the relationship between their weight and their hormones. For example, can you provide a little more detail about the women's responses to your interview question about hormones and provide clarity about which data were elicited from this question and which data have been interpreted by the researchers as effects of the women's weight loss on their hormones? My reason for asking this is because I find this sentence in the discussion section (line 27-30) quite confusing/ambiguous "Although hormonal changes after surgery were not a central focus, the women talked about related physical alterations that could be interpreted as hormonal effects that became a part of the whole experience." Did the women have any thoughts on how hormones affect their bodies/how weight loss had affected their hormones or were they unable to answer the question because they lacked knowledge understanding of how hormones affect their physical and psychological health and well-being – you allude to this in the discussion section and state in the conclusions that "changes in sex hormones were mentioned but not central in the women's narratives". I am not clear if the reason that sex hormones were not central is because they were of lesser importance/significance to the women or because they lacked knowledge on sex hormones to be able to articulate their feelings. If the women lacked sufficient knowledge about sex hormones, then this is a limitation that you should acknowledge more fully. You describe in the results section that at least some of the women were aware that there is a link between obesity and subfertility. I am also confused about whether your description of restitution of menstrual cycles refers to data derived from your interviews or from the quantitative studies that you mention earlier in the sentence because you do not discuss menstrual cycles in the results section.
--	---

VERSION 1 – AUTHOR RESPONSE

Reviewer: 1

Mrs. Catherine Homer, Sheffield Hallam University Comments to the Author:

The article includes fascinating quotes explaining the experiences of women living with severe obesity who have undergone bariatric surgery.

I was expecting the article to include more findings on the experiences of sex changes in hormones, given the emphasis on this in the introduction - which is seemingly what you wanted to discuss. However, I think the quotes and experiences of women are not related strongly enough to this to be such a major part and I wonder if there could be some refocus to be explicit about what the article does add, not what it doesn't. There are lots of really interesting themes that have come out that support other works published from across the world, and I feel some of this gets lost in what you may have hoped to say.

Some specific areas to consider below.

page 4, line 14 - 'The use of hormone-based contraceptives may have been a confounding factor influencing experiences in changes of hormones in some women.' I am not sure these features in the rest of the paper are enough to be brought into this section. it is very much the author's assumption as opposed to what is actually presented from the data. There is no other mention of the word contraceptives in the article, and I would expect more relevance to this statement to be included in the article summary.

R: We appreciate this input. We included this discussion of hormone-based contraceptives as a potential limitation to the study due to the potential of masking variations in the female hormone cycle leading to reduced experiences of hormones. Noted as Reviewer 1 comment 1.

Page 10, lines 3 - 15 - very long sentence. Consider rewording.

R: Thanks for this feedback, we agree. The sentence is shortened and revised. Noted as Reviewer 1 comment 2.

Page 19, line 10 - 'Findings can preferably be included in the information to women seeking bariatric surgery and be repeated as a routine during follow-up' - Unclear what this means - please consider rewording.

R: Thanks for noticing. We have clarified the meaning of this sentence. Noted as Reviewer 1 comment 3.

Reviewer: 2

Ms. Clare Robertson, University of Aberdeen Comments to the Author:

Thank you for the opportunity to read this interesting paper. The study provides valuable insights into the experiences of women who have undergone bariatric surgery. I have a few suggestions to hopefully improve the manuscript for a wider audience who may be less familiar with issues surrounding obesity and fertility/sex hormones and/or may have less experience with qualitative methods.

Introduction

Line 25: To improve the readability for a wider audience, it would be helpful to provide lay descriptions of some of the less well-known technical terms relating to fertility, such as fecundity, oligo/anovulation and oocytes – I think fertility, irregular periods and egg cells are possibly more commonly used terms?

R: Thank you for the suggestions to increase readability and provide information to a broader audience. We have changed accordingly in the text. Noted as Reviewer 2 comment 1.

Line 44: Can you describe the effects on women's physical and psychosocial health and well-being? I presume these are negative effects?

R: Thanks for the comments and correct interpretation. We have revised the sentence and elaborated on the descriptions in the study concerning the detrimental effects of obesity on women's health and well-being for clarification. Noted as Reviewer 2 comment 2.

Methods

Line 42: Please can you provide a brief description of Gadamer's hermeneutic analysis and why you believe this was the best qualitative methodological approach to adopt to give you the "richer and deeper understanding of the experiences described by the women included in the study". Providing a fuller description of your methodological approach might address some of my other queries regarding the term 'facts' and interviewer reflexivity.

R: Thank you for providing the opportunity to explain. Using a hermeneutic technique and fusion of text, interpreter and fact (theory-derived) horizons allowed us to make interpretations relevant to our own subjective and professional experiences of female obesity/bariatric surgery and alterations in sex hormones and quality of life.

Having the ability to merge a fact horizon allowed us to explore current research, including recent studies and theories concerning the phenomena that the women in the study described. We have included a sentence at the beginning of the methods section to better elaborate on the motivation for the hermeneutic technique, according to Gadamer. Noted as Reviewer 2 comment 3.

Line 49: Can you please explain the phrase "the current accepted fact of the time"? What facts are you referring to here, and from whose perspective are these facts determined?

R: Thank you for the feedback and inquiry. Findings based on hermeneutic inquiry reflect the current time and society that participants are living in. For example, experiences and eventual research related to obesity 200 years ago may not be comparable to experiences in the modern day and, therefore, irrelevant to our research question.

The facts referred to are studies related to phenomena described by the participants. For example, relevant studies concerning the quality of life in women after undergoing bariatric surgery were reviewed after participants described experiences of improved quality of life after undergoing surgery [1].

Similarly, the findings concerning "ghost fat" after rapid weight loss from bariatric surgery were consulted after participants described their difficulties adjusting to new body size and appearance [2]. We have included a descriptive sentence to improve the explanation concerning "currently accepted facts of the time". Noted as Reviewer 2 comment 4.

Line 10-13: Please describe the process by which women were excluded from participating in an interview due to their difficulty in understanding questions or expressing themselves. How was this criterion determined, and by whom?

R: Important aspect, and we appreciate your very relevant query concerning inclusion and exclusion. The authors determined inclusion and exclusion criteria together. Language forms the basis of expression and understanding in qualitative inquiry, and comprehending questions from the investigator is a requirement of the study method to ensure relevance in the findings. Participants are required to comprehend the language of the country where the study occurred to provide reliable data successfully.

Utilizing interpreters was determined to be a confounding risk since the language transfer between investigator and participant risked becoming distorted. Participants were recruited based on inclusion determinants such as age, sex and time interval of surgery. Information was presented in Swedish, which may have led those needing to comprehend the language to opt-out.

There is a risk that transference is decreased, and the richness of the data is lost by not including persons with varying backgrounds that cannot communicate in Swedish. We have included this in the limitations section.

We have included a sentence in the methods elucidating on excluding translation-based communication. Noted as Reviewer 2 comment 5.

Line 31-33: Please expand your description of your decision for determining that ten participants was the number deemed to be an appropriate sample size. Was the sample size pre-determined before recruitment/data collection began, or did you achieve data saturation with ten participants? If you achieved data saturation, please describe your method for determining data saturation.

R: Thank you for this input. However, to our understanding, data saturation is utilized in grounded theory and is not a standard method in the hermeneutic technique.

It was determined by RP and experienced qualitative researcher CB, after the ten completed interviews, that the data amount and detail were sufficient to carry out analysis according to the hermeneutic technique without further recruitment. The number of participants was also based on previous study material, according to Kvale, describing 15 +/- 10 participants as a reliable population group [3]. We have included a short text and citation in the methods section to improve our description of the population's total number and data amount. Noted as Reviewer 2 comment 6.

Analysis section: You state earlier in the methods section that your methodological approach allows consideration of the pre-understanding of the interpreter. Can you provide more details of whether the researcher critically examined their own role and potential for bias during data collection and analysis and, if this was done, how this was achieved in practice?

R: Thank you for the questions concerning this aspect of hermeneutic technique. A main motivating factor in choosing hermeneutics is the inclusion of investigator pre-understanding and "bias" in interpreting the findings. This allows for a more realistic exploration of the participant's experiences since all individuals carry their own personal histories, experiences and beliefs that influence their realities and perception of the world around them.

During analysis of the text, the investigator describes their interpretation of the text excerpts from the transcribed interviews through written text. These investigator interpretations are based on the investigator's own subjective opinions and professional experience. The investigator texts (horizons) are merged into the final fusions to increase detailed descriptions of the investigated phenomena. We have added a short text in the analysis description to clarify the method of presenting investigator pre-understanding. Noted as Reviewer 2 comment 7.

Line 56-57: Again, please explain what you mean by facts in the sentence "current facts about the existing meaningful text were presented (fact horizon)."

R: Thanks for the reflection. As previously described, "current facts" are relevant studies and theories regarding the phenomena described by the participants in text excerpts from their transcribed interviews. The researchers undertake a review of current research related to the described phenomena to increase the richness of the fusions that emerge from the analysis. We added a short description of this in the text concerning analysis. Noted as Reviewer 2 comment 8.

Patients and public involvement statement Line 50: Can you please cite some of the previous studies that you discuss here?

R: Thank you for the feedback. We derived experience from previous studies and have now cited them in the text. Noted as Reviewer 2 comment 9.

Results and discussion

Given the emphasis on sex hormones in your study design, it would be helpful to have a little more detail/clarity on the women's thoughts on the relationship between their weight and their hormones. For example, can you provide a little more detail about the women's responses to your interview question about hormones and provide clarity about which data were elicited from this question and which data have been interpreted by the researchers as effects of the women's weight loss on their hormones? My reason for asking this is because I find this sentence in the discussion section (lines 27-30) quite confusing/ambiguous "Although hormonal changes after surgery were not a central focus, the women talked about related physical alterations that could be interpreted as hormonal

effects that became a part of the whole experience." Did the women have any thoughts on how hormones affect their bodies/how weight loss had affected their hormones or were they unable to answer the question because they lacked knowledge or understanding of how hormones affect their physical and psychological health and well-being – you allude to this in the discussion section and state in the conclusions that "changes in sex hormones were mentioned but not central in the women's narratives". I am not clear if the reason that sex hormones were not central is because they were of lesser importance/significance to the women or because they lacked knowledge on sex hormones to be able to articulate their feelings. If the women lacked sufficient knowledge about sex hormones, then this is a limitation that you should acknowledge more fully. You describe in the results section that at least some of the women were aware that there is a link between obesity and subfertility.

R: Thanks for spotting this divergence in the results and discussion. We added text to reduce the ambiguity of our sentence concerning women's responses to questions concerning sex hormones. Our original study focused on women's experiences of changed sex hormones after undergoing bariatric surgery and subsequent weight loss. However, during interviews, and despite open questions involving hormones, the women focused primarily on difficulties in life prior to surgery as well as improvements in overall physical function and body image after undergoing weight loss, and these phenomena became the primary outcome of the interview study.

Several women did allude to restoring menstrual cycles, improving fertility and reducing male-pattern fat dispersion. However, they did not prioritize these discussions in the interview situation despite attempts at probing and looping techniques. We conclude that some women may not have clearly understood the relationship between restored sex hormones and certain physical improvements. There were women, however, that showed an awareness of the damaging effects of obesity on fertility, although they may not have clearly understood the involvement of sex hormones in this process. We have added clarification about this in the limitations and elaboration of this in the discussion. Noted as Reviewer 2 comment 10.

I am also confused about whether your description of restitution of menstrual cycles refers to data derived from your interviews or from the quantitative studies that you mention earlier in the sentence because you do not discuss menstrual cycles in the results section.

R: We appreciate the request for clarification as it also improves our paper. A few participants described restored menstrual cycles and reduced menstrual symptoms, and we felt this was relevant to lift in the discussion even if it was not a primary focus; we have now added text and quotes in the results as it should be there before being discussed. Noted as Reviewer 2 comment 11.